# Cytomegalovirus-Associated Gianotti-Crosti Syndrome in 28-Year-Old Immunocompetent Patient

**DOI:** 10.3390/pathogens11111338

**Published:** 2022-11-12

**Authors:** Florence Dupont, Aurélien Aubry, Jean-Philippe Lanoix, Baptiste Demey

**Affiliations:** 1Department of Infectious Diseases, Amiens University Medical Center, F-80000 Amiens, France; 2Department of Virology, Amiens University Medical Center, F-80000 Amiens, France; 3UR UPJV 4294, Agents Infectieux, Résistance et Chimiothérapie (AGIR), Centre Universitaire de Recherche en Santé, Université de Picardie Jules Verne, F-80000 Amiens, France

**Keywords:** Cytomegalovirus, Gianotti-Crosti, acrodermatitis, CMV

## Abstract

Gianotti-Crosti syndrome is a cutaneous eruption that occurs rarely in adults. It mostly concerns pediatric population and immunocompromised patients. Cytomegalovirus has already been described as one etiology of Gianotti-Crosti acrodermatitis in children and bone-marrow transplanted patients. Here, we present a Cytomegalovirus-associated Gianotti-Crosti syndrome in a 28-year-old immunocompetent female patient diagnosed in CHU Amiens-Picardie (Amiens, France). This type of case has never been shared in literature before. This rare complication of Cytomegalovirus infection indirectly led to disruption of anticoagulant treatment and thromboembolic incident that could have been fatal.

## 1. Introduction

Gianotti-Crosti syndrome (GCS) is a self-limiting cutaneous eruption characterized by papular eruption with a symmetrical distribution on the limbs and face [1]. It consists of monomorphic red-brown to pink papules and vesicles distributed symmetrically on the cheeks, extensor surface of the extremities, and buttocks [2]. The eruption usually spontaneously resolves over the course of 10 to 60 days. This syndrome frequently affects children aged 15 months to 2 years, but the etiology is rarely identified [3]. Some GCS case reports have been associated with viral pathogens such as hepatitis B virus, SARS-CoV-2, and Epstein-Barr virus [4,5,6]. It also could be subsequently observed after hepatitis B and Japanese encephalitis vaccination [7,8]. Despite being considered as an infantile disease, GCS can occur exceptionally in adult population [9], especially when immune system is weakened such as during HIV infection [10].

Another cause of GCS is the Cytomegalovirus (CMV) [3], a human herpesvirus responsible for a highly spread infection. This virus infects 70% of adults in industrialized countries and almost 100% in emerging countries, usually without apparent symptoms [11]. The infection can occur at each stage of the human life, from fetuses to elderly adults. Complications are more frequent in fragile patients, leading to congenital malformation if the fetus is infected or cytopathic and indirect organ diseases (retinitis, pneumonitis, hepatitis, gastrointestinal ulcerations, etc.) in immunocompromised patients. In immunocompetent population, the infection is usually asymptomatic or highlighted by hepatitis and mononucleosis syndrome [12].

CMV-associated GCS has already been described in children or bone marrow transplant recipients [13,14,15], but such case has never been reported in immunocompetent adults to our knowledge. Here, we aim to report a case of CMV-associated GCS in a 28-year-old immunocompetent female patient diagnosed in CHU Amiens-Picardie (Amiens, France). This rare complication of CMV infection indirectly led to the disruption of anticoagulant treatment and a thromboembolic incident.

## 2. Case Presentation

In 2022, a 28-year-old female patient went to the Emergency Department of CHU Amiens-Picardie (Amiens, France) for a 10-day history of fever associated with a rash. Her main history was thrombophilia due to factor V Leiden (treated with rivaroxaban) that lead to two previous pulmonary embolisms and sleep apnea syndrome. She has not traveled, except for a trip to Spain twelve years before, and has practiced unprotected sex several times since.

She presented fever, up to 40 °C, for 10 days, responding well to paracetamol. Concurrently, a macular rash appeared on the palms of the hands, elbows, and trunk. She also had a dry, emetogenic cough. This initial episode of hyperthermia was marked by a mild elevated C-reactive protein (CRP) level oscillating between 33 and 45 mg/L (normal value: <5 mg/L) without abnormality regarding to leucocyte counts and no evidence of mononucleosis syndrome on blood smear. The other laboratory tests were normal, including liver enzymes. Antibiotic therapy with Macrolides was initiated probably in the hypothesis of an atypical germ infection, for a total duration of 7 days. Due to the persistence of the fever and the rash after 3 days, she stopped the treatment, and she was referred to the Infectious Diseases Department of CHU Amiens-Picardie (Amiens, France) for an outpatient visit.

The dermatologic exam revealed maculo-papular lesions, without vesicle, on the extremities and the lower sides of the limbs (Figure 1). It was first hypothesized, jointly with the dermatologist, to be a Gianotti-Crosti papular acrodermatitis. The other hypotheses were a Coxsackie virus infection (but the lesions were not vesicular) and a secondary syphilis.

Screening for SARS-CoV-2, Hepatitis C virus, Epstein–Barr virus, and sexually transmitted agents (HIV, syphilis, *Chlamydia trachomatis*, and *Neisseria gonorrhoeae*) were negative. HBV serology showed a post-vaccination profile with 15.4 mIU/mL of anti-Hbs antibodies. Serology against CMV was performed by chemiluminescence on Alinity i^®^ automated system (Abbott Laboratories, Chicago, IL, USA). It revealed anti-CMV IgM with a signal equal to 4.71 relative light units (RLU) (positivity threshold: >1 RLU) and a 15 arbitrary units per milliliter (AU/mL) level of anti-CMV IgG (positivity threshold: 6 AU/mL). Anti-CMV IgG avidity assay, performed on the Vidas system (Biomérieux, Marcy I’Etoile, France), highlighted a low avidity, suggesting a recent CMV primary infection.

The diagnosis retained was therefore a primary CMV infection associated to a Gianotti-Crosti papular acrodermatitis. As the patient was immunocompetent, antiviral treatment was not introduced.

One week later, during the follow-up consultation, the patient expressed a disabling pain in the left calf. She was hospitalized for thrombosis exploration because of her medical history. Moreover, she reported non-compliance with her anticoagulant treatments and prolonged bed rest due to CMV infection and associated asthenia. Venous Doppler ultrasound revealed a deep vein thrombosis. The angiothoracic CT scan showed bilateral segmental embolism of the left upper lobe and the right lower lobe. During hospitalization, a PCR assay targeting CMV DNA in a plasma sample was realized on the COBAS 6800 system (Roche, Basel, Switzerland). At this time, the CMV viral load was 811 (2.91 log_10_) copies/mL. CRP levels were lower than previously but still elevated (9.8 mg/L), and no mononucleosis syndrome was perceptible on blood testing.

Thromboembolism was treated by one injection of tinzaparine (14000UI), then 15 mg of rivaroxaban twice a day for 21 days. 20 mg daily of rivaroxaban was prescribed after to prevent relapses.

One month after the symptom onset, the biological check-up showed a mononucleosis syndrome (leucocytes: 6160/mm^3^; lymphocytes: 3887/mm^3^ (63.1%); hyperbasophilic mononucleated cells), normal CRP and liver enzymes, ascending anti-CMV IgG (83.6 AU/mL), and still-detectable anti-CMV IgM. Dermatological lesions and thrombosis had completely disappeared.

## 3. Discussion

The present case report illustrates the variety of clinical manifestations of CMV primary infection. Cutaneous CMV is rarely reported in the literature, especially in immunocompetent adults, probably being underdiagnosed due to its non-specific clinical and biological features [16]. Indeed, in this case, CMV primary infection did not figure amongst the initial hypotheses, and the absence of mononucleosis biological syndrome can be misleading. Antiviral treatment is not recommended for immunocompetent patients, and as symptomatic treatment and rest is prescribed, doctors must stress compliance to other medication. Indeed, in our case, non-compliance led to thromboembolic events. To be noted, CMV infection may also be involved in the patient’s thrombotic episode. Indeed, the thromboembolic risk during acute CMV infection has already been widely described in the literature [17]. Transient antiphospholipid syndrome has been reported during infection [18]. Thrombosis may be associated with a procoagulant activity of infected monocytes [19], or with the virus itself via the expression of its envelope phospholipids [20]. Fortunately, the event was mild in this case.

In conclusion, we stress the importance to consider CMV primary infection in immunocompetent patients, even in front of dermatological lesions, and to insist on compliance in patients with co-morbidities and long-term drug treatment.

## Figures and Tables

**Figure 1 pathogens-11-01338-f001:**
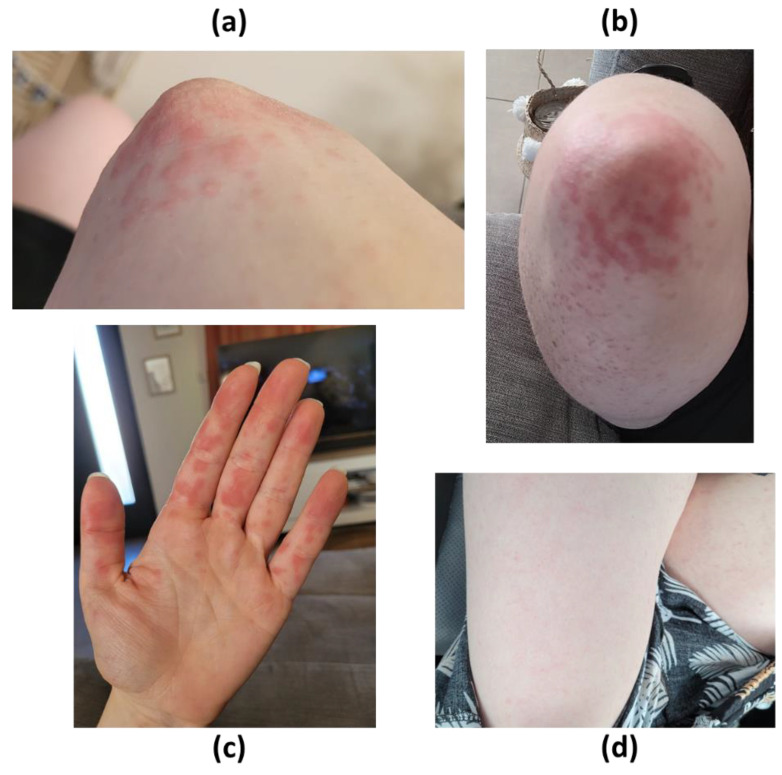
Dermatological lesions attributed to Cytomegalovirus-associated Gianotti-Crosti acrodermatitis. (**a**) Lateral view of right elbow. (**b**) Front view of right elbow. (**c**) Palm of left hand. (**d**) Front view of thighs.

## Data Availability

The data are not publicly available due to privacy and ethical restrictions.

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
