# Peer review of "Cytomegalovirus-Associated Gianotti-Crosti Syndrome in 28-Year-Old Immunocompetent Patient"

_pathogens, 2022, doi:10.3390/pathogens11111338_

Round 1
Reviewer 1 Report
This is an interesting case report on an adult immunocompetent patient presenting with Giannotti-Crosti syndrome associated with CMV. The case is well described and the cited literature is adequate. While this sydrome is common in young children it is rare or rarely described in otherwise healthy adults, thus of interest to the readers.
I have a few minor issues to be resolved:
1. Was hepatitis B tested in the patient? That is the classical association with Gianotti-Crosti syndrome.
2. In the introduction CMV is named a pandemic; this is misleading since it has been endemic for several centuries, please change or omit.
Author Response
Dear anonymous reviewer,
Thank you for the time you spent to make our manuscript better. It was edited according to your suggestions :
- Was hepatitis B tested in the patient? That is the classical association with Gianotti-Crosti syndrome.
--> The results for HBV testing were added as suggested
- In the introduction CMV is named a pandemic; this is misleading since it has been endemic for several centuries, please change or omit.
-->The term “pandemic” was removed to rephrase the sentence
Kind Regards,
Dr Baptiste Demey
Reviewer 2 Report
In this paper authors presented a case of immunocompetent adult with a Gianotti-Crosti syndrome as a consequence of CMV infection. The case is clearly presented and well documented. However, I suggest the authors to refer in the discussion to the possibility of an increased risk of thromboembolism due to the CMV infection itself, and not only as a consequence of not taking the prescribed therapy.
Author Response
Dear anonymous reviewer,
Thank you for the time you spent to make our manuscript better. It was edited according to your suggestions :
- I suggest the authors to refer in the discussion to the possibility of an increased risk of thromboembolism due to the CMV infection itself, and not only as a consequence of not taking the prescribed therapy.
--> A short paragraph was added in the discussion, as suggested.
Kind regards,
Dr Baptiste Demey